# Probiotic Formulations Containing Fixed and Essential Oils Ameliorates SIBO-Induced Gut Dysbiosis in Rats

**DOI:** 10.3390/ph16071041

**Published:** 2023-07-22

**Authors:** Ismail Aslan, Leyla Tarhan Celebi, Hulya Kayhan, Emine Kizilay, Mustafa Yavuz Gulbahar, Halil Kurt, Bekir Cakici

**Affiliations:** 1Department of Pharmaceutical Technology, Hamidiye Faculty of Pharmacy, University of Health Sciences, İstanbul 34668, Turkey; 2SFA R&D and Analysis Services, Teknopark İstanbul, İstanbul 34906, Turkey; leyla.tarhan@sfaarge.com (L.T.C.); bekir.cakici@sfaarge.com (B.C.); 3ATA BIO Technologies, Teknopol İstanbul, İstanbul 34930, Turkey; 4Art de Huile, Teknopol İstanbul, İstanbul 34930, Turkey; hulya.kayhan@artdehuile.com; 5Hamidiye Vocational School of Health Services, University of Health Sciences, İstanbul 34668, Turkey; emine.kizilay@sbu.edu.tr; 6Department of Pathology, Faculty of Veterinary Medicine, Ondokuz Mayıs University, Samsun 55200, Turkey; myg64@omu.edu.tr; 7Department of Medical Biology, Hamidiye International Faculty of Medicine, University of Health Sciences, İstanbul 34668, Turkey; halil.kurt@sbu.edu.tr

**Keywords:** dysbiosis, probiotic, small intestinal bacterial overgrowth (SIBO), essential oil

## Abstract

Dysbiosis of the gut microbiota is associated with the pathogenesis of intestinal diseases such as inflammatory bowel disease, irritable bowel syndrome (IBS), small intestinal bacterial overgrowth (SIBO), and metabolic disease states such as allergies, cardiovascular diseases, obesity, and diabetes. SIBO is a condition characterized by an increased number (>1 × 10^3^ CFU) of abnormal bacterial species in the small intestine. Interest in SIBO has gained importance due to increased awareness of the human microbiome and its potential relationships with human health and disease, which has encouraged new work in this area. In recent years, standard antibiotic regimens (rifaximin and metronidazole) have been used to treat SIBO, but solo antibiotics or their derivatives are insufficient. In this study, the therapeutic effects of the probiotic form, which contains coconut oil and traces of peppermint-lemon-patchouli essential oil, were evaluated on the Dysbiosis-Based Rat SIBO Model. There are significant differences between sick and healthy rats (*p* = 0.014), between sick rats and rats treated with the oil mix plus probiotic mix protocol (*p* = 0.026), and between rats treated with only the probiotic and only oil protocols (*p* = 0.030) in the evaluation of TNF-α levels. Histologically, villi distortion and loss of crypts, epithelial shedding and necrotic changes in the apical regions of the villi, and inflammatory cell infiltrations extending to the lamina propria and submucosa were observed in sick rats. Mitotic figures in villus epithelium and crypts were observed in rats treated with 9.2 × 10^9^ CFU/1000 mg/coconut oil + trace amounts of peppermint-lemon-patchouli essential oil and a probiotic mixture (oil + probiotic mix protocol). A regression of inflammatory reactions and an increase in goblet cells were observed. A decrease was observed in inflammation markers in sick rats. On the other hand, the oil plus probiotic mix protocol recovered digestive system defects in the animals caused by dysbiosis. In the future, these treatment approaches can be effective in the treatment of SIBO.

## 1. Introduction

The human gut is a complex ecosystem that contains billions of microorganisms, mostly colonized in the large intestine. Due to the antimicrobial effects of stomach acid and the peristaltic movement of the intestines, bacterial colonization in the small intestine is lower than in the large intestine. However, the stomach and proximal small intestine contain relatively few unique bacteria. Small intestinal bacterial overgrowth (SIBO), a sickness characterized by excessive bacterial colonization in the small intestine, can nevertheless emerge if normal homeostatic systems that regulate the host’s symbiotic bacterial count are disrupted. Protective factors that prevent this overgrowth in physiological conditions include stomach acid, intact anatomical integrity of the small intestine or mucosal integrity, normal small intestine motility (sweeping), a protective mucus layer, the enzymatic activities of pancreatic and bile secretions, and the protective effects of some beneficial bacteria. The etiologies of SIBO include anatomical changes, motility disorders, and gastric acid secretion abnormalities, as well as an imbalance of the microbiota that has been discovered in recent years. Therefore, irritable bowel syndrome (IBS), inflammatory bowel diseases (IBD), cirrhosis, gastric resection, gastroparesis, regular use of proton pump inhibitors, and recurrent antibiotic use are among the conditions that are risk factors for the formation of SIBO [1,2]. The presence of bacteria >10^5^ CFU/g-ml in the jejunal aspirate culture has traditionally been the accepted standard for diagnosis, but more recently, a bacterial concentration of >10^3^ CFU/mL has been accepted as a clinically relevant criterion for diagnosing SIBO [3]. However, the aspiration and culture method of small intestinal contents is an invasive and time-consuming procedure. In addition, this method lacks universal acceptance in terms of threshold values for diagnosing SIBO. As a result of this, breath tests—relatively rapid, straightforward, and indirect diagnostic procedures—have essentially replaced cultures in clinical practice for diagnosis of SIBO [3]. The hydrogen breath test (HBT) is a test method based on the detection of hydrogen released from the fermentation of carbohydrates by intestinal bacteria. SIBO can cause gastrointestinal disorders, malabsorption, bacterial translocation, and ultimately absorption problems. Bacterial overgrowth in the small intestine is often associated with digestive symptoms such as bloating, indigestion, diarrhea, and weight loss. Due to the small intestinal villus atrophy, important structural degenerations such as malabsorption and vitamin deficiencies occur [1]. SIBO has begun to attract attention as it is responsible for irritable bowel syndrome and liver damage, as well as vitamin, protein, and fat absorption disorders. Animal and human studies have shown the reversal and prevention of steatohepatitis following metronidazole treatment for SIBO. Recent studies have revealed that 65–85% of irritable bowel syndrome patients have SIBO [4]. Specifically, it has been suggested that the microbiota associated with the rectal mucosa (decreased relative aerobic, Gram-negative bacteria, and increased facultative anaerobic, Gram-positive bacteria) may function as a potential predictor of SIBO in IBS patients [5].

In recent years, rifaximin and metronidazole have been used as standard antibiotic regimens in the treatment of SIBO [6]. Antibiotic treatment of SIBO aims to improve symptoms of dysbiosis, potentially ameliorating other consequences of altered mucosal or luminal bacterial colonization. However, excessive use of antibiotics can worsen the intestinal ecosystem by disrupting the intestinal microflora. Therefore, long-term use of antibiotics indirectly causes dysbiosis. However, there are only a few studies in the literature reporting the efficacy of antibiotic treatment on bacterial overgrowth, and the results mention that antibiotics alone are insufficient for SIBO decontamination.

As a result of increased awareness about the potential relationships of the human microbiome with human health and diseases, interest in SIBO has increased in studies on this subject. Probiotics are an effective and appealing approach in the prevention of different diseases [7]. Recent studies have revealed promising results showing that probiotics can be an effective tool for both the prevention and treatment of SIBO. The important role of probiotics in the treatment of *Clostridium difficile* infection, infectious diarrhea, and irritable bowel syndrome has been proven by recent studies [8,9,10,11,12]. Orally administered lactic acid bacteria (Gram-positive *Lactobacillus* spp. and *Bifidobacterium* spp.) have been shown to reduce IBS symptoms by inhibiting the growth of enteric-type SIBO bacteria [5].

Probiotics prevent the dysbiosis of the gut microbiota that occurs after antibiotic use and fight off opportunistic pathogens. Therefore, probiotics may have significant advantages compared to antibiotics [8]. In general treatment approaches, it has been shown that the treatment effect is strengthened by antibiotic regimens supported by probiotics (decontamination rate: 85.8%), and the combined use of probiotics and antibiotics is promising for the treatment of SIBO [9].

Hypothetically, with a treatment regimen consisting of a combination of antibiotics and probiotics, pathogenic bacteria in the small intestine can be destroyed, toxins cleaned, and the protective gut microbiota rebuilt.

Coconut (*Cocos nucifera*) oil contains monolaurin, an antimicrobial and anti-inflammatory monoglyceride composed of lauric acid, a short-chain fatty acid that can disrupt the structure of the membranes of microbial organisms [13]. The antimicrobial effect of coconut oil on the skin has been proven to inhibit the proliferation of common skin pathogens such as *Propionibacterium acnes* and *Staphylococcus aureus* [14]. Another study demonstrated the potential of applying coconut oil to positively affect the stool microbiome with a significant increase in the abundance of probiotic bacteria such as *Lactobacillus*, *Allobaculum*, and *Bifidobacterium* species in the intestines [15]. Recent studies with various animal models have shown that lauric acid has low antimicrobial activity against commensal lactic acid bacteria in the human gut but high antimicrobial activity against *E. coli* and *Clostridium* [15]. Alongside this, another study with *Lactobacillus* species used in aromatic oils showed that the combined use of aromatic oils and probiotic bacteria suppressed enteric pathogens with a synergistic effect against intestinal infections [16].

In this study, it was aimed to investigate the therapeutic effects of probiotic bacteria combined with aromatic fixed and essential oils on SIBO using the dysbiosis-based SIBO animal model.

## 2. Results

### 2.1. Microbiome Analysis

The results of the analysis were evaluated by expressing the graphs in which the bacterial ratios were compared. A total of 19,211,765 readings were obtained from 100 samples, and the mean readings of the samples were determined as 192,118 ± 98,647. After the readings were cleaned with Dada2, an average of 23,335 ± 18,593 clean readings were obtained and used in the next steps. According to the alpha rarefaction graph, sufficient reading depth was achieved in all samples. Alpha diversity analysis was used to assess the variation in microbial diversity between groups. The results showed that fecal microbial alpha diversity was significantly increased in rats treated with the probiotic mix pl oil mix protocol (oil_plus_probiotic) compared to other groups.

In contrast, the microbial diversity was reduced in the sick rats (Figure 1). According to the statistical comparison of Shannon from the alpha diversity index (Figure 1f), a statistically significant difference was found between healthy rats and rats treated with the probiotic mix plus oil mix protocol and between sick rats and rats treated with the probiotic mix plus oil mix protocol (respectively *p* value 0026, 0.0002). In the same index, a statistically significant difference was found between the rats to which only the probiotic mix protocol was applied and the rats to which the oil mix + probiotic mix protocol was applied, and between the rats to which only the oil mix protocol was applied and the rats to which the oil mix + probiotic mix protocol was applied (*p* values are 0.0016 and 0.014, respectively).

According to the Venn bacteria diagram, a total of 117 of 822 OTUs were shared by five groups (Figure 1). Therefore, these 117 OTUs constitute the core microbiota. While 81 OTUs were observed only in the healthy group, the numbers of OTUs observed only in the sick, probiotic, oil, and fat plus probiotic groups were 65, 46, 75, and 63, respectively.

When beta diversity between samples is examined, a significantly different distribution is detected between the groups (Figure 2). These results show that the group that received the oil-probiotic mix protocol differed significantly from the other groups. Significant differences in beta diversity were observed among all groups (*p* = 0.001). Significant beta diversity was found between sick and healthy rats; between sick rats and rats that only took the oil mix protocol; and between healthy rats and rats that took the oil mix plus probiotic mix protocol (*p* values are 0.023, 0.002, and 0.004, respectively).

According to the 16S rRNA gene LEfSe analysis (see Section 4.11 Statistical Analysis) performed to identify differentially abundant taxa in the groups, 53 bacterial taxa showed significant relative abundance between the groups (Figure 3). In our samples, *Actinobacteriota*, *Bacteroides*, and *Firmicutes* were the most commonly appearing phyla (Figure 4). The frequency of *Proteobacteria* and *Firmicutes* phyla was higher in the sick group than in the other groups (Figure 4). The relative frequencies of the samples are given as bar graphs at the level of phylum and family (Figure 5).

### 2.2. Biochemical Analysis Results

Proinflammatory factor analyses were performed on healthy and sick rat groups, and the groups treated with treatment protocols were evaluated with the *t*-test for different variances. Accordingly, there was a significant difference in IL-1β levels (Figure 6) between healthy and sick rats and between sick rats and rats treated with an oil mix plus probiotic mix (*p* < 0.05). There was a significant difference in IL-6 levels (Figure 6) between healthy and sick rats and between sick rats and rats treated with the oil mix plus probiotic mix protocol (*p* < 0.05). There was no significant difference between healthy rats and rats treated with the oil mix plus probiotic mix protocol (*p* = 0.754). There were significant differences between sick and healthy rats (*p* = 0.014), between sick rats and rats treated with the oil mix plus probiotic mix protocol (*p* = 0.026), and between rats treated with only probiotic and only oil protocols (*p* = 0.030) in the evaluation of TNF-α levels. Also, a significant difference was found between the rats treated with only oil and the rats administered the oil mix plus probiotic mix protocol (*p* = 0.023). On the other hand, there were no significant differences between rats treated with only the probiotic mix protocol and rats treated with the oil mix plus probiotic mix protocol (*p* = 0.315) or between healthy rats and rats treated with the oil mix plus probiotic mix protocol (*p* = 0.681) according to TNF-α level (Figure 6).

### 2.3. Histopathological Analysis

Epithelial structure and infiltration scoring were performed in the tissues examined histopathologically [17,18]. There was a significant difference between the rats that received only the oil mix protocol and the rats that received only the probiotic mix protocol and the rats that received only the oil mix protocol, and the rats that used the oil + probiotic mix protocol (*p* < 0.05). There was no significant difference (*p* = 0.241) in inflammation score between the rats that were administered only the probiotic mix protocol and the rats that were administered the oil plus probiotic mix protocol (Figure 7).

When the differences observed between the healthy group and the other four groups were examined by veterinary physical examination, abdominal bloating and flatulence on palpation and agitation were observed in the sick rats compared to the healthy rats. After sacrification, inflammatory thickenings and necrotic areas were found among the macroscopic findings of the digestive system of the sick rats (Figure 8). In the histopathological examination performed in the sick rat group, villous distortion, loss of crypts and goblet cells, epithelial shedding in the apicals of the villus, and inflammatory cell infiltrations extending to the lamina propria and submucosa were observed (Figure 8). In the examination of the rat group that received only the oil mix protocol, a decrease in inflammatory cell infiltrations and a decrease in submucosal edema were observed in thick and coarsened villi (Figure 8). In the examination of the rat group that received only the probiotic mix protocol, fusion of the villi and a decrease in mononuclear cell infiltration in the lamina proria and submucosa were detected. At the same time, an increase in mitotic activity and regenerative hyperchromatic epithelial cells was observed in the crypts (Figure 8). On the other hand, in rats treated with the oil plus probiotic mix protocol, inflammatory reactions decreased or disappeared (as an indicator of the repair phase); many mitotic figures in the villi epithelium and crypts were observed, along with an increase in goblet cells and fusion of the villi in areas with regenerated epithelium (Figure 9).

## 3. Discussion

This study consisted of five different groups of rats treated with three different treatment regimens. A total of 50 rats were included in the study. When the parameters discussed in this study are evaluated, it is hopeful that the biochemical, histopathological, and microbiological criteria show parallelism and support the diagnosis and treatment pathways, meaning that the study has achieved its purpose. However, the “Dysbiosis-Based Rat SIBO Model” [19] study provides evidence that probiotic mixtures are more effective when administered in oil. Decreased alpha and beta diversity (Figure 1) and an increase in inflammation markers (Figure 6) were also reflected in histopathological scoring (Table 1 and Figure 7) in patient rats with a confirmed diagnosis of SIBO. Accordingly, in the patient rat group in which alpha and beta diversity were impaired, the increase in both diversities was correlated histologically and biochemically, especially in the group in which the oil + probe protocol was applied after the application of the treatment protocols. Inflammation detected in scoring was also confirmed by proinflammatory cytokine levels. Elevated IL-1β, IL-6, and TNF-α values in sick rats (Figure 6 and Table 2) decreased to healthy levels in rats treated with the oil mix plus probiotic mix protocol. This regression was visualized histologically (Figure 9 and Figure 10). Improvement in inflammation scoring is clearly demonstrated (Table 1 and Figure 7).

The potent anti-inflammatory effects of some strains of probiotics that have gained attention in recent years have clearly highlighted how the therapeutic potential of these agents may go beyond their ability to displace other organisms, leading to considerations of their potential use in inflammatory bowel disease. When McCarthy et al. [20,21] measured the effects of the colonization of both *Lactobacillus* and *Bifidobacterium* in the colon and cecum, it was observed that the levels of interferon (IFN), tumor necrosis factor (TNF), and interleukin (IL)-12 remained within the control values. Similar effects have been shown for probiotic cocktails in experimental models of colitis. Other studies have demonstrated the ability of probiotics not only to interfere with pathogen adhesion and invasion but also to neutralize bacterial toxins and improve the mucosal barrier function [7]. Wagner et al. examined the effects of probiotic supplementation on gastrointestinal symptoms and SIBO after “Gastric Bypass” in their prospective, randomized, double-blind, placebo-controlled studies and found that *Lactobacillus acidophilus* and *Bifidobacterium lactis* supplements were effective in reducing bloating [11]. García-Collinot et al. suggested that 2-month *S. boulardii* treatment reduced the complaints of diarrhea, abdominal gas, and abdominal pain in SIBO patients with systemic sclerosis [16]. In the clinical study performed in this patient cohort, it was found that monotherapy with *S. boulardii* was more effective than metronidazole in eliminating SIBO and improving its associated symptoms; however, it was observed that the best results were obtained when the combination of metronidazole and *S. boulardii* was combined [22]. In a meta-analyis and systematic review prepared by 18 studies, the findings demonstrated that probiotic supplementation significantly decontaminates SIBO, alleviates abdominal pain, and lowers H_2_ concentration [9]. In vivo animal studies and clinical trials demonstrate the key roles probiotics play in human gut microbiome-associated diseases. It has been proved by many clinical trials that probiotics could shape the intestinal microbiota due to their potential for controlling multiple bowel diseases and promotion of overall wellness [23].

In our study, it was observed that the short-chain inulin and probiotic mix content applied to sick rats with a confirmed diagnosis of SIBO gave significant results in repair. According to the results obtained, probiotics in oil replacement have an anti-inflammatory effect on proinflammatory cytokines (IL-1β, IL-6, TNF-α). In addition, it has been determined that the form here under study acts as a regulator (see Section 2.2 biochemical analysis results). A significant difference between healthy rats and sick rats is a supporting finding in terms of dysbiosis and inflammation. On the other hand, the fact that the proinflammatory cytokines of the sick rats that received the protocol (oil mix + probiotic mix) came back to a healthy range is promising in that it will regulate the immune response developing in many diseases. A significant correlation was seen between histopathological analyses and analyses of proinflammatory cytokines. Considering the results of inflammation scoring and proinflammatory cytokines, it was observed that the probiotic mixed form in coconut oil and trace amounts of peppermint-lemon-patchouli essential oils corrected dysbiosis and inflammation, provided repair in the duodenum, and provided findings for the treatment of SIBO. However, today, many conditions affect the effectiveness of probiotics. There are problems such as completely damaged microbiota and reduced microbiome diversity in traumatized intestines. However, when the results of existing studies with probiotics are interpreted, differences are observed in terms of patient demographics, study design, age of enrollment, type, variety, dose, and duration of use of probiotics [24]. Finally, numerous confounding factors in the host can alter the therapeutic effect of probiotics, including the composition of the diet, the drugs used, the intestinal transit time, and the amount of gastric acid secretion [3]. However, studies have shown that probiotic supplementation is an effective option for SIBO decontamination and the relief of abdominal swelling and pain. In this study, behavioral disorders such as abdominal distension, flatulence, agitation, and defecation pattern (diarrhea, loose stools, or constipation) were detected on palpation [25,26], and microbiome analyses of rats were correlated with physical findings. In the microbiome analysis results (Figure 3), increases in pathogenic flora (*Escherichia* spp. and *Clostridium* spp.) and changes in alpha and beta diversity were in the direction of dysbiosis compared to the healthy control group, supporting the diagnosis of SIBO. Upon the statistical evaluation of the data obtained from the groups to which treatment protocols were applied, a significant increase in the alpha-beta diversity was observed in the groups in which the probiotic form in the oil mixture was applied, as opposed to only probiotic replacement or only oil mixture replacement. In the future, these treatment approaches can be effective in the treatment of SIBO.

## 4. Materials and Methods

### 4.1. Test Animals

In this study, 25 female and 25 male *Sprague-Dawley* adult rats (12-month-old) were used. The experimental groups consisted of female rats with an average weight of 275 g and male rats with an average weight of 310 g. The cages were cleaned every morning. The average consumption of feed was 10–20 g per rat (DSA RT01), and water was 8–10 mL per day. The feeding of the rats and all other experimental studies were carried out following the “TUBITAK Animal Experiments Local Ethics Committee Directive”.

### 4.2. Experimental Design

The healthy group was separated for the study, and a special nutritional diet was applied to the experimental groups to create dysbiosis in the large intestine and SIBO in the small intestine. Antibiotic agents that would shift the intestinal microbiota in one direction and a combination of chemical agents (polyacrylamide) and trans-fat-containing foods were given, and the microbiota of the large intestine was disrupted (dysbiosis). Dietary material containing amoxicillin, clavulanic acid, and macrolide (to impair motility activity for SIBO) was administered at approximately 200 mg/kg/day by gastric gavage (oral to stomach tube) for 4 weeks. Long-term use of antibiotics generally leads to dysbiosis. In particular, the “macrolide” group (clarithromycin) of antibiotics impairs motility and reverses the sweeping movement from the small intestine to the large intestine. Impaired gut motility leads to the migration of microorganisms from the large intestine microbiota to the small intestine, and therefore, by causing a more than expected increase in microorganisms in the small intestine, it has prepared the environment for SIBO [27]. To confirm SIBO formation and stimulate inflammation in rats, trans fat (100 mg/kg/day) was added to the diet, and 1 mL/day of 10% bicarbonate water was added to increase gastric pH, and a proton pump inhibitor of 3 mg/kg/day for 4 weeks was applied. Before each feeding, a physical examination of the rats was performed to observe significant bloating and gas. Rats with gastrointestinal swelling and increased gas were sacrificed, and histopathological groaning was performed. After the diagnosis of the disease was confirmed histologically, the sick rats were divided into different treatment groups, and the study continued (Figure 10). 10 rats were used for each group.

### 4.3. Rat Study Groups

A total of 50 adult rats were divided into 5 groups, each consisting of 10 rats (*n* = 10), as below:

Group 1: This was left as the healthy rat group. This group was not given a diet to induce inflammation or dysbiosis. They were fed normally in a separate room from the other rat groups.

Group 2: The rats were left sick in this group. After the diagnosis of the disease was confirmed, no external application was made without feeding.

Group 3: After the diagnosis of the disease was confirmed, only the oil mix protocol (1 g/day of coconut oil plus a trace amount of peppermint-lemon-patchouli essential oil) was applied.

Group 4: After the diagnosis of the disease was confirmed, only the probiotic mix (1 × 10^10^ CFU/1000 mg/day probiotic + 3 mg/day short-chain inulin mixed in water) protocol was applied.

Group 5: After the diagnosis of the disease was confirmed, the oil mix + probiotic mix (1 g/day, 1 × 10^10^ CFU/1000 mg/day coconut oil + probiotic mix with trace amounts of peppermint-lemon-patchouli essential oil) protocol was applied.

### 4.4. Application Protocol

Normal feeding protocol: The first group of rats left as the healthy group was fed normally (10–20 g feed, 8–10 mL water per daily average rat) in a separate room from the other rat groups.

Sick rat feeding protocol: This is the application to sick rats after the diagnosis of SIBO. Only normal feeding was applied.

Only oil mix protocol: This is the application to the third group after the diagnosis of SIBO. Trace amounts of essential medicinal peppermint, patchouli, and lemon oil in coconut oil were given to the rats by gastric gavage every day (1 mL/day) for 3 weeks (Table 3).

Only Probiotic Mix Protocol: This is the application to the fourth group after the diagnosis of SIBO. A mixture of bacteria (*Lactobacillus rhamnosus* ATA-LRS1902, *Lactobacillus gastricus* ATA-LRG01101, *Lactobacillus acidophilus* ATA-LAP1201, *Bifidobacterium longum* ATA-BSP1908, *Bifidobacterium bifidum* ATA-BSP1709, *Bifidobacterium animalis* ATA-BSLA0310, *Lactobacillus plantarum* ATA-LPC98052, *Bacillus clausii* LTCS-BSSL01140, *Akkermansia muciniphila* ATA-MDA013081, *Bacillus subtilis* LTCS-BSP031101) dissolved in distilled water containing 3 mg of inulin was given to the rats by gastric gavage every day (1 mL/day) for 3 weeks (Table 4).

Probiotic mix + oil mix protocol: It is the application to the fifth group after the diagnosis of SIBO. A mixture of bacteria (*Lactobacillus rhamnosus* ATA-LRS1902, *Lactobacillus gastricus* ATA-LRG01101, *Lactobacillus acidophilus* ATA-LAP1201, *Bifidobacterium longum* ATA-BSP1908, *Bifidobacterium bifidum* ATA-BSP1709, *Bifidobacterium animalis* ATA-BSLA0310, *Lactobacillus plantarum* ATA-LPC98052, *Bacillus clausii* LTCS-BSSL01140, *Akkermansia muciniphila* ATA-MDA013081, and *Bacillus subtilis* LTCS-BSP031101) in a mixture of fixed and essential oils (coconut oil, lemon, patchouli, and medicinal peppermint essential oil) was given daily for 3 weeks (1 mL/day) by gastric gavage (Table 5).

### 4.5. Collection and Storage of Samples

Fecal samples were collected directly into a falcon tube (without falling into the cage) during cage cleaning for healthy rats, while other groups were fed by gastric gavage directly into the falcon tube. The fecal collection was done daily in groups. Feces were also collected from the cage, but samples directly from experimental animals and samples from the ileocecal portion after sacrification were used for analysis. Collected feces were stored at −80 °C for microbiome analysis. For sacrification, subcutaneous anesthesia (1 mL of ketamine hydrochloride) was applied and left for about 5 min. Other sampling procedures were performed after the fainting.

For biochemical analyses, left ventricular blood was taken and placed in a gel tube during sacrification. The gel tubes were centrifuged at 3000 RPM for 5 min, and the sera were separated. Separated sera were frozen at −80 °C for biochemical analyses (Interleukin 1 and 6, TNF-α) and stored until analysis. Tissues taken after sacrification were fixed in 10% buffered formalin and stored at +4 °C for histopathological examinations. A total of 50 rats were included in the study, and 100 feces preparations for microbiome analysis, 39 serum preparations for biochemical analysis, and 50 duodenal section preparations for histopathological analysis were examined. During the studies, 11 blood samples, 2 from the first group, 1 from the second group, 3 from the third group, 3 from the fourth group, and 2 from the fifth group, could not be included in the analysis because of hemolysis.

### 4.6. Diagnostic Analysis Steps

For inflammation, dysbiosis, and SIBO, a physical examination was performed by veterinarians after an aggressive diet for 4 weeks, and the significant differences observed between the healthy group and the other four groups were evaluated. For this reason, in comparison with healthy rats, behavioral disorders such as abdominal bloating, flatulence, agitation on palpation, and defecation pattern (diarrhea, loose stools, or constipation) were examined [25,26]. To confirm the physical examination diagnosis, one rat was sacrificed, and the correlation of tissue or organ examinations with disease markers and physical findings was investigated. In the microbiota analysis, changes in putrifying (corrosive) flora and acidic flora, changes in microbiota, increases in pathogenic flora, histopathological changes, and basic proinflammatory cytokines (IL-1β, IL-6, TNFα) were examined compared to healthy rats [25,28], and results were evaluated statistically.

### 4.7. Fecal Gene Extraction

Feces taken from the rats divided into groups before the start of the applications, during the applications, at the end of the applications, and during the sacrification were analyzed. Microbiome analysis was performed on rat stools divided into groups before and after the applications. Genetic material extraction was performed from stool samples using the QIAamp DNA Stool Mini Kit (Qiagen, Hilden, Germany) by the manufacturers’ instructions. Genomic material concentration and quality were measured using NanoDrop spectrophotometry (Nanodrop, Erlangen, Germany).

### 4.8. High-Throughput 16S rRNA Gene Sequence and Data Analysis

16S rRNA gene, primer set targeting V3-V4 region amplified by the PCR method

(F:5′-TCGTCGGCAGCGTCAGATGTGTATAAGAGACAGCCTACGGGNGGCWGCAG-′3. R: 5′GTCTCGTGGGCTCGGAGATGTGTATAAGAGACAGGACTACHVGGGTATCTAATCC-′3.).

The obtained amplicons were purified using the illumina Nextera XT index kit. After the library was created, sequencing was done via the sequencing by synthesis method. The data produced after sequencing was converted to raw data (FASTQ format) and made ready for analysis. Raw reads were analyzed using QIIME2 version 2021.11 [29]. After the reads were cleaned with the Dada2 command, taxonomic assignments were made using the Silva v1.38 (https://www.arb-silva.de, accessed on 1 May 2023) database.

### 4.9. Biochemical Analysis

Rat sera collected throughout the study were Elabscience Rat IL-1β (Interleukin 1 Beta) (Catalog No: E-EL-R0012-96T), BTL Rat IL-6 (Cat. No E0135Ra), and BTL TNF-α (Cat. No E0764Ra). The sera were tested using kits working on the Sandwich-ELISA principle. The micro-ELISA plate included in the kit had been pre-coated with an antibody specific to rat IL-1β. Standards and samples were added to the micro-ELISA plate wells and conjugated with the specific antibody. Optical density (OD) was measured spectrophotometrically at a wavelength of 450 nm ± 2 nm. The OD value was evaluated in proportion to the rat IL-1β, IL-6, and TNF-α concentrations [19,28,30].

### 4.10. Histopathological Analysis

When the tissues were taken after sacrification and fixed in 10% buffered formalin, a routine tissue follow-up was performed and embedded in paraffin blocks. Sections of 5 µm from the paraffin blocks were stained with hematoxylin-eosin [22]. The preparations were examined under a light microscope (LM = Nikon Eclipse E600, Minato, Japan), and the lesions were scored [17,18]. In scoring, crypt structure, epithelial changes in the apical regions of villi, necrotic-mitotic changes, inflammatory cell changes in lamina propria, submucosa, or villi, submucosal edema, hyperplastic changes, and goblet cell changes were examined. At the end of the examination, epithelial structure and infiltration (inflammation) scoring was done.

### 4.11. Statistical Analysis

Statistical analysis included the following: alpha and beta diversity indices qiime2 diversity plugin to evaluate the change of microbial diversity between groups; PERMANOVA test to evaluate the effects of different phenotypes between groups; unpaired Student *t*-test; and different variances for calculation (*p* values) in comparisons between groups. The analysis was done with a two-way *t*-test. The species relationship between the groups was evaluated with the Venn Diagram. For alpha diversity, the difference between the groups was investigated according to the statistical comparison of the Shannon Diversity Index, and a *p*-value of <0.05 was considered statistically significant. Baseline coordinate analysis (PCoA) was performed using the “unweighted UniFrac” method to monitor beta variation between groups. Results were evaluated using permutation multivariate analysis of variance (PERMANOVA), and *p* < 0.05 was considered statistically significant. However, in the evaluation of biochemical analyses, a *t*-test between different variances was used, and the *p* < 0.05 value was considered statistically significant. To identify differentially abundant taxa in the groups, linear-discriminant analysis coupled with effect-size analysis (LEfSe) was performed on fecal microbiota composition between groups, based on 16S rRNA gene sequencing, and an LDA score > 3 was considered statistically significant.

## 5. Conclusions

This study observed that the important role of probiotics in ameliorating SIBO is further enhanced by their combination with oil. Villi distortion and loss of crypts, epithelial shedding and necrotic changes in the apical regions of the villi, and inflammatory cell infiltrations extending to the lamina propria and submucosa were observed in sick rats. Mitotic figures in the villus epithelium and crypts were observed in rats treated with 9.2 × 10^9^ CFU/1000 mg/coconut oil + trace amounts of peppermint-lemon-patchouli essential oil and a probiotic mixture (oil + probiotic mix protocol), with regression of inflammatory reactions and an increase in goblet cells. A decrease was observed in inflammation markers in sick rats. On the other hand, the rats treated with the oil plus probiotic mix protocol recovered from digestive system defects caused by dysbiosis. Reproducing these results with well-designed prospective clinical trials will pave the way for the introduction of probiotics into routine clinical use in SIBO.

## Figures and Tables

**Figure 1 pharmaceuticals-16-01041-f001:**
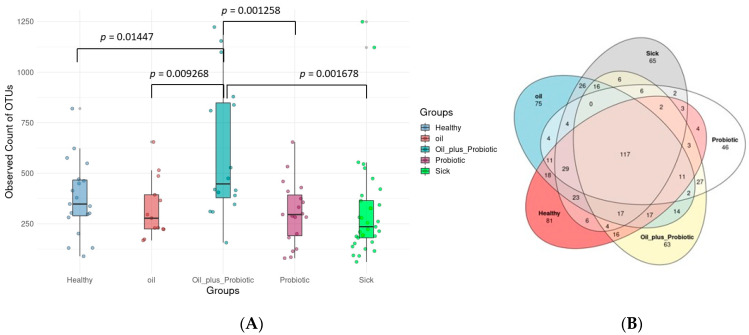
(**A**) Fecal microbial diversity is estimated by the Shannon index. *n* = 100 (*n* = 19 for healthy rats; *n* = 33 for sick rats; *n* = 19 for rats in which only probiotic mix protocol was applied; *n* = 13 for rats in which only oil mix protocol was applied; *n* = 13 for rats that were applied oil mix + probiotic mix protocol = 16). The *p*-values were calculated with the bilateral unpaired Student’s *t*-test. Bars represent the standard deviation. (**B**) Overlaps between groups in the Venn diagram *n* = 100 (*n* = 19 for healthy rats; *n* = 33 for sick rats; *n* = 19 for rats in which only probiotic mix protocol was applied; *n* = 13 for rats in which only oil mix protocol was applied; *n* = 13 for rats that were applied oil mix + probiotic mix protocol = 16).

**Figure 2 pharmaceuticals-16-01041-f002:**
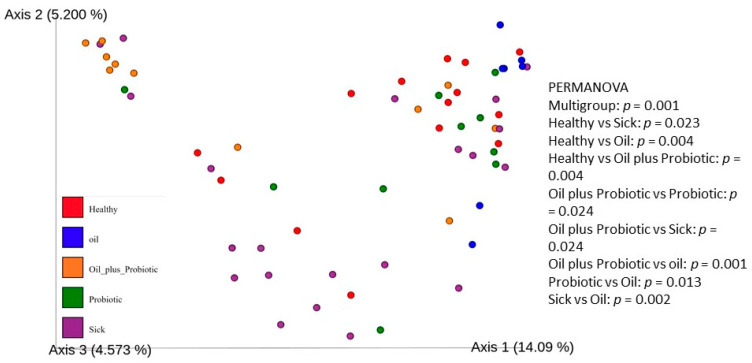
PCoA of unweighted UniFrac distances and beta diversity was calculated by PERMANOVA. Permutation multivariate analysis of PERMANOVA analysis of variance.

**Figure 3 pharmaceuticals-16-01041-f003:**
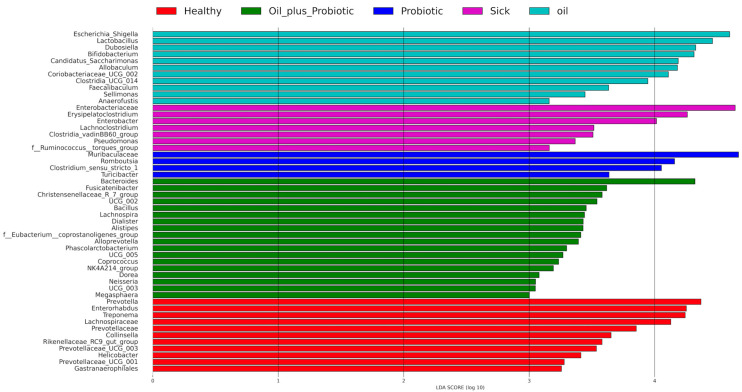
LDA histogram combined with effective size measurement based on 16S rRNA gene sequence logarithmic LDA score > 3.0.

**Figure 4 pharmaceuticals-16-01041-f004:**
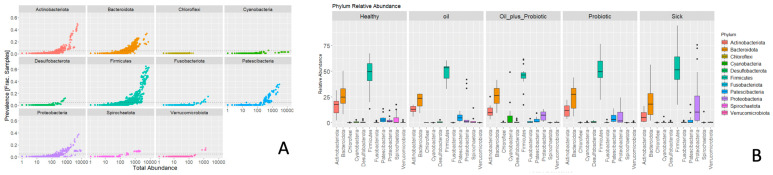
(**A**) The most common visible phyla. (**B**) Graph of relative abundance of groups at the phylum level.

**Figure 5 pharmaceuticals-16-01041-f005:**
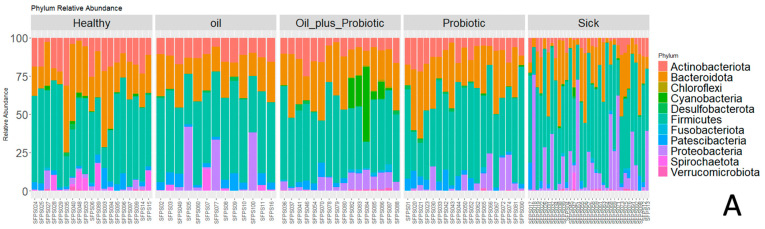
(**A**) Bar graphs of the samples at the phylum level. (**B**) Family-level bar graphs of the samples.

**Figure 6 pharmaceuticals-16-01041-f006:**
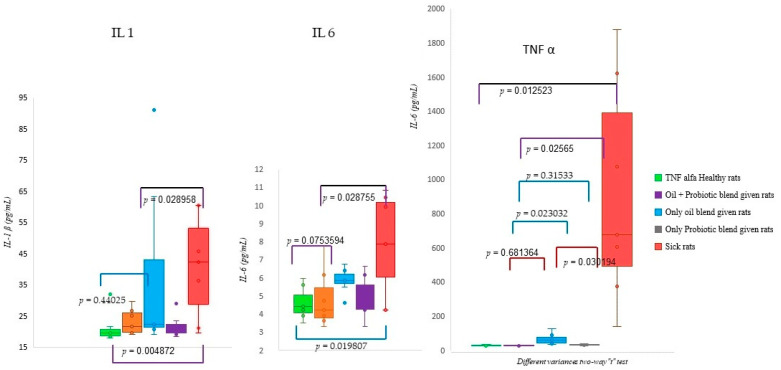
IL-1β, IL-6, and TNF-α results; the graph of difference was evaluated with *t*-tests for two-way differences. Healthy rats, *n* = 8; rats given oil mix plus probiotic mix, *n* = 7; rats given only oil mix, *n* = 8; rats given only Probiotic mix, *n* = 7; sick rats, *n* = 9.

**Figure 7 pharmaceuticals-16-01041-f007:**
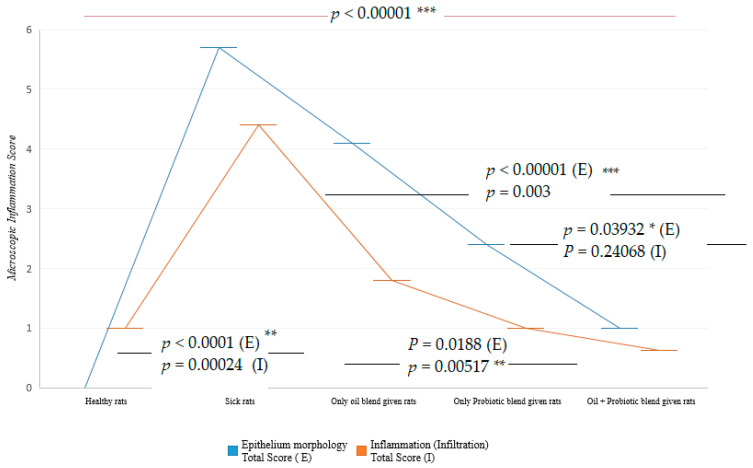
Epithelial morphology and infiltration scoring, different variances, two-way *t*-test graph. P (E): Epithelial score; P (I): Inflammation score (* *p* < 0.05, ** *p* < 0.01, *** *p* < 0.001).

**Figure 8 pharmaceuticals-16-01041-f008:**
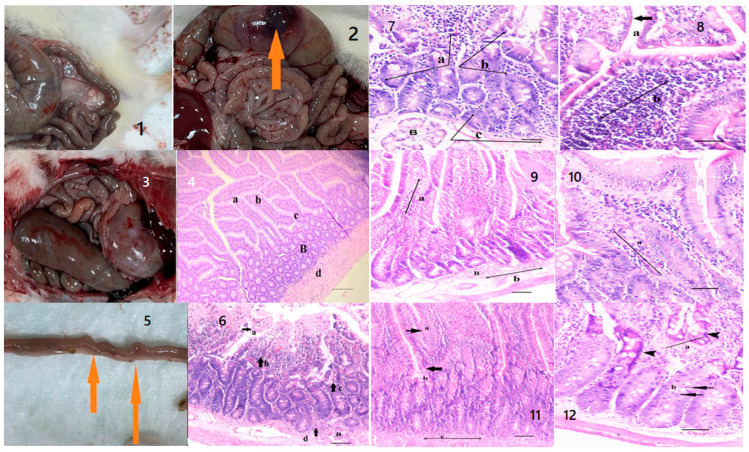
Healthy (**1**) and sick (**2**) rats. Necrotic area (arrow) in the wall of the large intestine. (**3**) Healthy rat, macroscopic view after sacrification. (**4**) Section of healthy duodenum: Goblet cells (a), Lamina propria (c), Crypt cells (b), Submucosa (d), healthy Brunner’s glands. (B), Duodenum, HE, bar = 45 μm. (**5**) Sick rat, thickening of the duodenum due to inflammatory infiltration (arrows). (**6**) villus distortion and villi epithelial shedding in their apical regions (a), loss in crypts (c), lamina inflammatory cell infiltrates extending into the propria (b) and submucosa (d), disrupted Brunner’s glands (B) Duodenum, HE, bar = 45 μm. (**7**) Loss of crypts (c) and goblet cells (b), lamina in the propria mononuclear cell infiltration (a), Brunner’s Glands (B). (**8**) thickened villi with loss of goblet cells in epithelial layer thinned areas in the epithelium (a thick arrow), lamina dense mononuclear cell infiltrates in the propria (b, thin arrow). Duodenum, HE, Bar = 20 μm. (**9**) Decreased inflammatory cell infiltrates in thick and coarsened villi (a), decreased submucosal edema. Brunner’s glands (B). (**10**) lamina inflammatory cells in the propria (a). Duodenum HE, Bar = 20 μm. (**11**) Villi, some of which are coarse and thick, sometimes fused with each other(a), lamina reduced mononuclear cell infiltration in the propria (b) and submucosa (c). Duodenum, HE, bar = 90 μm. (**12**) increased mitotic activity in crypts (arrows, (a)) alongside regenerative hyperchromatic epithelial cells (arrowheads, (b)). Duodenum HE, Bar = 20 μm.

**Figure 9 pharmaceuticals-16-01041-f009:**
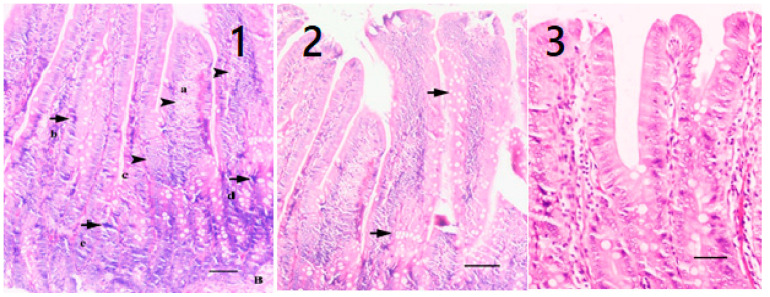
(**1**) Decreased inflammatory reaction; villi: numerous mitotic figures (arrowheads **a**–**e**) in their epithelium and crypts. Brunner’s glands (B). (**2**) elongated and hyperplastic in places in epithelial villi increase in goblet cells (arrows), Duodenum, HE, bar = 45 μm. (**3**) regenerated sometimes fused villi in the region of epithelium. There is no inflammatory reaction. Duodenum, HE, bar = 20 μm.

**Figure 10 pharmaceuticals-16-01041-f010:**
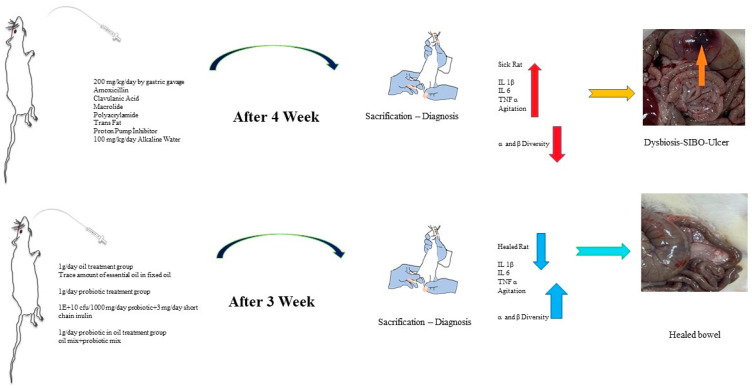
The schematic illustration of the study design.

**Table 1 pharmaceuticals-16-01041-t001:** Epithelial morphology and infiltration score.

Group		Healthy Rats Score Average (*n* = 10)	Sick Rats Score Average (*n* = 10)	Only Oil Mix Given Rats Score Average (*n* = 10)	Only Probiotic Mix Given Rats Score Average (*n* = 10)	Oil + Probiotic Mix Given Rats Score Average (*n* = 8)
Epithelium Morphology	Normal morphology (0)	0	0	0	0	0
	Loss of goblet cells (Low) (1)	1	1	1	1	1
	Loss of goblet cells in large areas (Strong) (2)	0	2	0	0	0
	Loss of crypts (Low) (3)	0	3	3	3	0
	Loss of crypts in large areas (Strong) (4)	0	4	4	4	0
Inflammation	No infiltrates (0)	0	0	0	0	0
	Infiltrate around crypt basis (1)	1	0	1	1	1
	Infiltrate reaching to Lamina muscularis mucosae (2)	0	0	2	2	0
	Extensive infiltration reaching the Lamina muscularis mucosae and thickening of mucosa with edema (3)	0	3	0	0	0
	Infiltration of the Lamina Submucosa (4)	0	4	0	0	0

**Table 2 pharmaceuticals-16-01041-t002:** Mean values of inflammation marker.

Mean Value	IL-1β (pg/mL)	IL-6 (pg/mL)	TNF-α (pg/mL)
Healthy rats	20.87	6.13	35.73
Oil mix + Probiotic mix given rats	22.84	4.82	37.47
Only oil mix given rats	37.30	5.84	73.55
Only Probiotic mix given rats	21.57	4.92	39.25
Sick rats	34.30	7.56	847.54

**Table 3 pharmaceuticals-16-01041-t003:** Oil mix content and quantities.

Component Name	Amount of Mg
Coconut oil	985 mg/mL
Medicinal peppermint oil	5 mg/mL
Patchouli oil	5 mg/mL
Lemon oil	5 mg/mL
Total	1000 mg/mL

**Table 4 pharmaceuticals-16-01041-t004:** Bacterial mix content and quantities.

Component Name	Amount of Mg	Amount of CFU
Pure water	967 mg/mL	-
Inulin	3 mg/mL	-
*Lactobacillus rhamnosus* ATA-LRS1902	5 mg/mL	2.00 × 10^9^ CFU/mL
*Lactobacillus gastricus* ATA-LRG01101	5 mg/mL	1.00 × 10^9^ CFU/mL
*Lactobacillus acidophilus* ATA-LAP1201	4 mg/mL	1.00 × 10^9^ CFU/mL
*Bifidobacterium longum* ATA-BSP1908	4 mg/mL	1.00 × 10^9^ CFU/mL
*Bifidobacterium bifidum* ATA-BSP1709	3 mg/mL	1.00 × 10^9^ CFU/mL
*Bifidobacterium animalis* ATA-BSLA0310	3 mg/mL	1.00 × 10^9^ CFU/mL
*Lactobacillus plantarum* ATA-LPC98052	3 mg/mL	1.00 × 10^9^ CFU/mL
*Bacillus clausii* LTCS-BSSL01140	1 mg/mL	1.00 × 10^9^ CFU/mL
*Akkermansia muciniphila* ATA-MDA013081	1 mg/mL	1.00 × 10^8^ CFU/mL
*Bacillus subtilis* LTCS-BSP031101	1 mg/mL	1.00 × 10^8^ CFU/mL
Total	1000 mg/mL	9.20 × 10^9^ CFU/mL

**Table 5 pharmaceuticals-16-01041-t005:** Bacteria mixture in oil mixture content and quantities.

Component Name	Amount of Mg	Amount of CFU
Coconut oil	955 mg/mL	-
Medicinal Peppermint oil	5 mg/mL	-
Patchouli oil	5 mg/mL	-
Lemon oil	5 mg/mL	-
*Lactobacillus rhamnosus* ATA-LRS1902	5 mg/mL	2.00 × 10^9^ CFU/mL
*Lactobacillus gastricus* ATA-LRG01101	5 mg/mL	1.00 × 10^9^ CFU/mL
*Lactobacillus acidophilus* ATA-LAP1201	4 mg/mL	1.00 × 10^9^ CFU/mL
*Bifidobacterium longum* ATA-BSP1908	4 mg/mL	1.00 × 10^9^ CFU/mL
*Bifidobacterium bifidum* ATA-BSP1709	3 mg/mL	1.00 × 10^9^ CFU/mL
*Bifidobacterium animalis* ATA-BSLA0310	3 mg/mL	1.00 × 10^9^ CFU/mL
*Lactobacillus plantarum* ATA-LPC98052	3 mg/mL	1.00 × 10^9^ CFU/mL
*Bacillus clausii* LTCS-BSSL01140	1 mg/mL	1.00 × 10^9^ CFU/mL
*Akkermansia muciniphila* ATA-MDA013081	1 mg/mL	1.00 × 10^8^ CFU/mL
*Bacillus subtilis* LTCS-BSP031101	1 mg/mL	1.00 × 10^8^ CFU/mL
Total	1000 mg/mL	9.20 × 10^9^ CFU/mL

## Data Availability

Data is contained within the article.

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
