# Peer review of "Probiotic Formulations Containing Fixed and Essential Oils Ameliorates SIBO-Induced Gut Dysbiosis in Rats"

_pharmaceuticals, 2023, doi:10.3390/ph16071041_

Round 1

Reviewer 1 Report

Comments

The present study reported that the therapeutic effect of probiotic form containing Coconut oil and trace amounts of Peppermint-Lemon-Patchouli essential oil on SIBO. Study is interesting however, below mentioned points as well native person needs to polish English version.

Line 19-22: Rephrase the sentence “There is increasing evidence that dysbiosis of the gut microbiota is associated with the pathogenesis of intestinal diseases such as inflammatory bowel disease, irritable bowel syndrome (IBS) and small intestinal bacterial overgrowth (SIBO), and metabolic disease states such as allergies, cardiovascular diseases, and obesity and diabetes”.

Line 38: In keywords what is the abbreviation of SIBO?

Line 84-86: However, there are only a few studies in the literature reporting the efficacy of antibiotic treatment of bacterial overgrowth and antibiotics are insufficient for SIBO decontamination, alone rephrase the sentence and add the recent references.

Line 88: Add sentence “Probiotics are effective and appealing approach in the prevention of different diseases” and add the recent references such as https://doi.org/10.1080/00439339.2021.1883412  and 10.29261/pakvetj/2021.00

Line 122-149: Clearly mentioned the number of experimental animal groups with their replication

Line 151: Check the picture clearly following journal format

Line 194-195: Check the table clearly following journal format and some words are italic correct them

Line 451: Here S. boulardii should be italic

Line 454: Here S. Boulardii should be italic

Line 444: Wagner et al. examined the effects of probiotic supplementation on gastrointestinal symptoms and SIBO after “Gastric Bypass” here correct the reference following journal standard format

Line 447: Here B.lactis should be italic and check all over the manuscript

Line 500-502: What the future perspectives researchers can take following your trial? need more details

Comment: Need to check and improve figures, diagrams, and references following the journal format

Native person needs to polish English version.

Author Response

Reviewer 1

The present study reported that the therapeutic effect of probiotic form containing Coconut oil and trace amounts of Peppermint-Lemon-Patchouli essential oil on SIBO.

  1. Study is interesting however, below mentioned points as well native person needs to polish English version.

Response: Firstly, I really thank you so much for your supportive revisions. The manuscript was read by a native researcher.

  1. Line 19-22: Rephrase the sentence “There is increasing evidence that dysbiosis of the gut microbiota is associated with the pathogenesis of intestinal diseases such as inflammatory bowel disease, irritable bowel syndrome (IBS) and small intestinal bacterial overgrowth (SIBO), and metabolic disease states such as allergies, cardiovascular diseases, and obesity and diabetes”.

Response: It was revised according to the reviewer’s suggestion.

  1. Line 38: In keywords what is the abbreviation of SIBO?

Response: It was revised according to the reviewer’s suggestion.

  1. Line 84-86: However, there are only a few studies in the literature reporting the efficacy of antibiotic treatment of bacterial overgrowth and antibiotics are insufficient for SIBO decontamination, alone rephrase the sentence and add the recent references.

Response: It was revised according to the reviewer’s suggestion.

  1. Line 88: Add sentence “Probiotics are effective and appealing approach in the prevention of different diseases” and add the recent references such as https://doi.org/10.1080/00439339.2021.1883412  and 10.29261/pakvetj/2021.00

Response: The sentence was added according to the reviewer’s suggestion.

  1. Line 122-149: Clearly mentioned the number of experimental animal groups with their replication

Response: It was revised according to the reviewer’s suggestion.

  1. Line 151: Check the picture clearly following journal format

Response: It was revised according to the journal format.

  1. Line 194-195: Check the table clearly following journal format and some words are italic correct them

Response: It was revised according to the journal format.

  1. Line 451: Here S. boulardii should be italic

Response: It was corrected.

  1. Line 454: Here S. Boulardii should be italic

Response: It was corrected.

  1. Line 444: Wagner et al. examined the effects of probiotic supplementation on gastrointestinal symptoms and SIBO after “Gastric Bypass” here correct the reference following journal standard format

Response: It was revised according to the journal format.

  1. Line 447: Here B.lactis should be italic and check all over the manuscript

Response: It was corrected.

  1. Line 500-502: What the future perspectives researchers can take following your trial? need more details

Response: A sentence was added according to the reviewer’s suggestion.

  1. Comment: Need to check and improve figures, diagrams, and references following the journal format

Response: All checked according to the journal format.

Reviewer 2 Report

The authors aimed to evaluate the symbiotic (synergistic) effect of a probiotic mix (P+EO; Table 2, 9.2 x 109 CFU.mL-1) + essential oil mix [coconut (98.5%)/peppermint-lemon-patchouli (1.5%)] in rats with antibiotic/polyacrylamide/trans-fat-induced enteral dysbiosis [small intestinal bacterial overgrowth (SIBO)]. The anti-inflammatory potential (Histology/interleukins) and microbiota features (16S rRNA Meta sequencing). Differences in beta diversity (PERMNOVA, p≤0.024) and genus level (LEfSe analysis) were detected among groups and anti-inflammatory features where group-specific in a healthy>P+EO>control groups (P, EO alone)>untreated rats’ trend. Minor changes could further improve the study´s scientific soundness: 

Title. Very long and with abbreviations. Suggestion: A probiotic consortium + essential oils, ameliorates antibiotic-induced gut dysbiosis in rats. 

Abstract. Please describe more quantitatively than qualitatively (include p-values)

Introduction/methods. OK

Tables & Figures. A) All tables and figures (increase resolution >300 dpi) should be supplied according to the journal´s guidelines. B) Please consider providing them some of them as supplementary material. C) Figure 4: chloroplast? mitochondria? D) “The most common visible bacteria” are not visible, E) Significant figures (p-values) should not have more than three decimal places. F) Table 5 should describe significant differences. G) Provide enough details as footnotes when needed.

Discussion. Even though the probiotic + essential oil formulation used in this study is unique, the authors should make a more comparative discussion of the results obtained compared to other reports using a similar dysbiosis model and where symbiotic formulations are used to counteract it.

References. A) Review again the citation format according to the instructions for the authors. B) Reduce the ratio of new/old references (<10y) to say 25%.

Extensive editing of English language required

Author Response

Reviewer 2

The authors aimed to evaluate the symbiotic (synergistic) effect of a probiotic mix (P+EO; Table 2, 9.2 x 109 CFU.mL-1) + essential oil mix [coconut (98.5%)/peppermint-lemon-patchouli (1.5%)] in rats with antibiotic/polyacrylamide/trans-fat-induced enteral dysbiosis [small intestinal bacterial overgrowth (SIBO)]. The anti-inflammatory potential (Histology/interleukins) and microbiota features (16S rRNA Meta sequencing). Differences in beta diversity (PERMNOVA, p≤0.024) and genus level (LEfSe analysis) were detected among groups and anti-inflammatory features where group-specific in a healthy>P+EO>control groups (P, EO alone)>untreated rats’ trend. Minor changes could further improve the study´s scientific soundness: 

  1. Title. Very long and with abbreviations. Suggestion: A probiotic consortium + essential oils, ameliorates antibiotic-induced gut dysbiosis in rats. 

Response: Firstly, I really thank you so much for your supportive revisions. The title was revised according to the reviewer’s suggestion.

  1. Abstract. Please describe more quantitatively than qualitatively (include p-values)

Response: It was revised according to the reviewer’s suggestion.

  1. A) All tables and figures (increase resolution >300 dpi) should be supplied according to the journal´s guidelines.

Response: It was revised according to the journal’s format.

  1. B) Please consider providing them some of them as supplementary material.

Response: Thank the reviewer for his/her suggestion.

  1. C) Figure 4: chloroplast? mitochondria?

Response: It was revised according to the reviewer’s suggestion.

  1. D) “The most common visible bacteria” are not visible,

Response: It was revised according to the reviewer’s suggestion.

  1. E) Significant figures (p-values) should not have more than three decimal places.

Response: It was revised according to the reviewer’s suggestion.

  1. F) Table 5 should describe significant differences.

Response: The differences in significance were given in the results part.

  1. G) Provide enough details as footnotes when needed.

Response: It was revised according to the reviewer’s suggestion.

  1. Discussion. Even though the probiotic + essential oil formulation used in this study is unique, the authors should make a more comparative discussion of the results obtained compared to other reports using a similar dysbiosis model and where symbiotic formulations are used to counteract it.

References. A) Review again the citation format according to the instructions for the authors. B) Reduce the ratio of new/old references (<10y) to say 25%.

Reviewer 3 Report

I reviewed the manuscript titled “The Therapeutic Effect of Probiotic Form Containing Coconut Oil and Trace Amounts of Peppermint-Lemon-Patchouli Essential Oil on SIBO.

Please remove the colon (: punctuation mark) at the end of the title

Line 29: authors can remove the full name as it is already abbreviated elsewhere in the manuscript

Conclusions and recommendations of the study must be introduced in abstract

Introduction

Research gaps should be addressed with more recent literature. Please add more recent literature

Figure 1 quality must be improved. The test is, as such, is not readable

Scientific names must be in Italics

The section results are missing

Authors should keep the word

Most of the figures quality must be improved

The results presented are appropriate

The discussion section must be improved. As such, it is weak. I suggest authors to add more in-depth discussion on research findings and its comparison with available literature

References must be aligned with journal format 

Author Response

Reviewer 3

I reviewed the manuscript titled “The Therapeutic Effect of Probiotic Form Containing Coconut Oil and Trace Amounts of Peppermint-Lemon-Patchouli Essential Oil on SIBO.

  1. Please remove the colon (: punctuation mark) at the end of the title

Response: Firstly, I really thank you so much for your supportive revisions. It was corrected.

  1. Line 29: authors can remove the full name as it is already abbreviated elsewhere in the manuscript

Response: It was revised according to the reviewer’s suggestion.

  1. Conclusions and recommendations of the study must be introduced in abstract

Response: It was revised according to the reviewer’s suggestion.

  1. Research gaps should be addressed with more recent literature. Please add more recent literature

Response: It was revised according to the reviewer’s suggestion.

  1. Figure 1 quality must be improved. The test is, as such, is not readable

Response: The figures were revised according to the reviewer’s suggestion.

  1. Scientific names must be in Italics

Response: It was revised according to the reviewer’s suggestion.

  1. The section results are missing

Response: It was revised according to the reviewer’s suggestion.

  1. Most of the figures quality must be improved

Response: The figures were revised according to the reviewer’s suggestion.

  1. The discussion section must be improved. As such, it is weak. I suggest authors to add more in-depth discussion on research findings and its comparison with the available literature

Response: The figures were revised according to the reviewer’s suggestion.

  1. References must be aligned with journal format 

Response: It was revised according to the journal’s format.

Round 2

Reviewer 1 Report

Check the references carefully

Minor English editing is required

Reviewer 2 Report

Thanks for having accepted most of my suggestions

minor changes